# An Efficient Transfer Learning Framework for Multiagent Reinforcement Learning

**Tianpei Yang**[1],[*] **Weixun Wang**[1],[*] **Hongyao Tang**[1],[*] **Jianye Hao**[12],[†] **Zhaopeng Meng**[1],
**Hangyu Mao**[2], **Dong Li**[2], **Wulong Liu**[2], **Chengwei Zhang**[3], **Yujing Hu**[4],
**Yingfeng Chen**[4], **Changjie Fan**[4]

[1]College of Intelligence and Computing, Tianjin University
{tpyang,wxwang,bluecontra,jianye.hao,mengzp}@tju.edu.cn
[2]Noah's Ark Lab, Huawei, {maohangyu1,lidong106,liuwulong}@huawei.com
[3]Dalian Maritime University, chenvy@dlmu.edu.cn
[4]NetEase Fuxi AI Lab, {huyujing, chenyingfeng1, fanchangjie}@corp.netease.com

## Abstract

Transfer Learning has shown great potential to enhance single-agent Reinforcement Learning (RL) efficiency. Similarly, Multiagent RL (MARL) can also be accelerated if agents can share knowledge with each other. However, it remains a problem of how an agent should learn from other agents. In this paper, we propose a novel Multiagent Policy Transfer Framework (MAPTF) to improve MARL efficiency. MAPTF learns which agent's policy is the best to reuse for each agent and when to terminate it by modeling multiagent policy transfer as the option learning problem. Furthermore, in practice, the option module can only collect all agent's local experiences for update due to the partial observability of the environment. While in this setting, each agent's experience may be inconsistent with each other, which may cause the inaccuracy and oscillation of the option-value's estimation. Therefore, we propose a novel option learning algorithm, the successor representation option learning to solve it by decoupling the environment dynamics from rewards and learning the option-value under each agent's preference. MAPTF can be easily combined with existing deep RL and MARL approaches, and experimental results show it significantly boosts the performance of existing methods in both discrete and continuous state spaces.

## 1 Introduction

Transfer Learning has achieved expressive success of accelerating single-agent Reinforcement Learning (RL) via leveraging prior knowledge from past learned policies of relevant tasks [37, 36]. Inspired by this, transfer learning in Multiagent Reinforcement Learning (MARL) [6, 17, 4, 16, 7, 8] is also studied with two major directions: 1) transferring knowledge across different but similar tasks and 2) transferring knowledge among multiple agents in the same task. The former usually explicitly computes similarities between tasks [18, 3, 10] to transfer across similar tasks with the same number of agents, or design network structures to transfer across tasks with different numbers of agents [1, 33]. In this paper, we focus on the latter due to the following intuition: in a Multiagent System (MAS), each agent's experience is different, so the states each agent visits (the familiarities to different regions of the environment) are also different; if we figure out some principled ways to transfer knowledge across different agents, all agents could form a big picture even without exploring the whole space of the environment, which will facilitate more efficient MARL.

---

[*]Equal contribution. † Corresponding author.

35th Conference on Neural Information Processing Systems (NeurIPS 2021).

In fact, the latter direction is still investigated at an initial stage, and the assumptions and designs of some recent methods are usually simple. For example, LeCTR [27] and HMAT [19] adopt the teacher-student framework to enable each agent to learn when to advise other agents or receive advice from other agents. However, they only consider a two-agent scenario. Later, PAT [22] extends this idea to scenarios with more than two agents, and enables each agent to learn from other agents through an attentional teacher selector. However, it simply uses the difference of two unbounded value functions as the student reward which may cause instability.

DVM [32] and LTCR [35] are two methods to transfer knowledge among multiple agents through policy distillation. However, they simply decompose the training process into two stages (i.e., the learning phase and the transfer phase) by turns, which is a coarse-grained manner. Moreover, they consider the equal significance of knowledge transfer throughout the whole training process, which is counter-intuitive. A good transfer should be adaptive rather than being equally treated, e.g., the transfer should be more frequent at the beginning of the training since agents are less knowledgeable about the environment, while decay as the training process continues because agents are familiar with the environment gradually and should focus more on their own experiences.

In this paper, we propose a novel Multiagent Policy Transfer Framework (MAPTF) which models the policy transfer among multiple agents as the option learning problem. In contrast to previous teacher-student and policy distillation frameworks, MAPTF is adaptive and applicable to scenarios consisting of more than two agents. Specifically, MAPTF adaptively selects a suitable policy for each agent to exploit, which is imitated by an agent as a complementary optimization objective. MAPTF also uses the termination probability as a performance indicator to determine whether the exploitation should be terminated to avoid negative transfer. Furthermore, due to partial observability of the environment, the update of the option-value function is based on all agent's local experience. However, in this setting, each agent's experience may be inconsistent, which could cause the option-value estimation to oscillate and become inaccurate. A novel option learning algorithm, the Successor Representation Option (SRO) learning is used to overcome this inconsistency by decoupling environment dynamics from rewards to learn the option-value function under each agent's preference. MAPTF can be easily incorporated into existing Deep RL and MARL approaches. Our simulations show it significantly boosts the performance of existing approaches both in discrete and continuous state spaces.

## 2 Preliminaries

**Partially Observable Stochastic Games.** Stochastic Games [24] are a natural multiagent extension of Markov Decision Processes (MDPs), which model the dynamic interactions among multiple agents. Considering the fact agents may not have access to the complete environmental information, we follow previous work's settings and model the multiagent learning problems as partially observable stochastic games [13]. A Partially Observable Stochastic Game (POSG) is defined as a tuple $\langle \mathcal{N}, \mathcal{S}, \mathcal{A}^1, \cdots, \mathcal{A}^n, \mathcal{T}, \mathcal{R}^1, \cdots, \mathcal{R}^n, \mathcal{O}^1, \cdots, \mathcal{O}^n \rangle$, where $\mathcal{N}$ is the set of agents; $\mathcal{S}$ is the set of states; $\mathcal{A}^i$ is the set of actions available to agent $i$ (the joint action space $\mathcal{A} = \mathcal{A}^1 \times \mathcal{A}^2 \times \cdots \times \mathcal{A}^n$); $\mathcal{T}$ is the transition function that defines transition probabilities between global states: $\mathcal{S} \times \mathcal{A} \times \mathcal{S} \to [0, 1]$; $\mathcal{R}^i$ is the reward function for agent $i$: $\mathcal{S} \times \mathcal{A} \to \mathbb{R}$ and $\mathcal{O}^i$ is the set of observations for agent $i$. A policy $\pi^i$: $\mathcal{O}^i \times \mathcal{A}^i \to [0, 1]$ specifies the probability distribution over the action space of agent $i$. The goal of agent $i$ is to learn a policy $\pi^i$ that maximizes the expected return with a discount factor $\gamma$: $J = \mathbb{E}_{\pi^i} \left[ \sum_{t=0}^{\infty} \gamma^t r_t^i \right]$.

**The Options Framework.** Sutton et al. [30] firstly formalized the idea of temporally extended action as an option. An option $\omega \in \Omega$ is defined as a triple $\{\mathcal{I}^\omega, \pi^\omega, \beta^\omega\}$ in which $\mathcal{I}^\omega \subset \mathcal{S}$ is an initiation state set, $\pi^\omega$ is an intra-option policy and $\beta^\omega : \mathcal{I}^\omega \to [0, 1]$ is a termination function that specifies the probability an option $\omega$ terminates at state $s \in \mathcal{I}^\omega$. An MDP endowed with a set of options becomes a Semi-MDP, which has a corresponding optimal option-value function over options learned using intra-option learning. The options framework considers the *call-and-return* option execution model, in which an agent picks an option $\omega$ according to its option-value function $Q_\omega(s, \omega)$, and follows the intra-option policy $\pi^\omega$ until termination, then selects a next option and repeats the procedure.

**Deep Successor Representation (DSR).** The successor representation (SR) [9] is a basic scheme that describes the state value function by a prediction about the future occurrence of all states under a fixed policy. SR decouples the dynamics of the environment from the rewards. Given a transition

$(s, a, s', r)$, SR is defined as the expected discounted future state occupancy:

$$M(s, s', a) = \mathbb{E}\left[\sum_{t=0}^{\infty} \gamma^t \mathbb{1}[s_t = s'] | s_0 = s, a_0 = a\right], \tag{1}$$

where $\mathbb{1}[.]$ is an indicator function with value of one when the argument is true and zero otherwise. Given the SR, the Q-value for selecting action $a$ at state $s$ can be formulated as the inner product of the SR and the immediate reward: $Q^\pi(s, a) = \sum_{s' \in \mathcal{S}} M(s, s', a)\mathcal{R}(s')$.

DSR [21] extends SR by approximating it using neural networks. Specifically, each state $s$ is represented by a feature vector $\phi_s$, which is the output of the network parameterized by $\theta$. Given $\phi_s$, SR is represented as $m_{sr}(\phi_s, a|\tau)$ parameterized by $\tau$, a decoder $g_{\bar{\theta}}(\phi_s)$ parameterized by $\bar{\theta}$ outputs the input reconstruction $\hat{s}$, and the immediate reward at state $s$ is approximated as a linear function of $\phi_s$: $\mathcal{R}(s) \approx \phi_s \cdot \mathbf{w}$, where $\mathbf{w} \in \mathbb{R}^D$ is the weight vector. In this way, the Q-value function can be approximated by putting these two parts together as: $Q^\pi(s, a) \approx m_{sr}(\phi_s, a|\tau) \cdot \mathbf{w}$. The stochastic gradient descent is used to update parameters $(\theta, \tau, \mathbf{w}, \bar{\theta})$. Specifically, the loss function of $\tau$ is:

$$L(\tau, \theta) = \mathbb{E}\left[(\phi_s + \gamma m'_{sr}(\phi_{s'}, a'|\tau') - m_{sr}(\phi_s, a|\tau))^2\right], \tag{2}$$

where $a' = \mathrm{argmax}_a\, m_{sr}(\phi'_s, a) \cdot \mathbf{w}$, and $m'_{sr}$ is the target SR network parameterized by $\tau'$ which follows DQN [26] for stable training. The reward weight $\mathbf{w}$ is updated by minimizing the loss function: $L(\mathbf{w}, \theta) = (\mathcal{R}(s) - \phi_s \cdot \mathbf{w})^2$. The parameter $\bar{\theta}$ is updated using an L2 loss: $L(\bar{\theta}, \theta) = (\hat{s} - s)^2$. Thus, the loss function of DSR is the composition of the three loss functions: $L(\theta, \tau, \mathbf{w}, \bar{\theta}) = L(\tau, \theta) + L(\mathbf{w}, \theta) + L(\bar{\theta}, \theta)$.

## 3 Multiagent Policy Transfer Framework (MAPTF)

### 3.1 Framework Overview

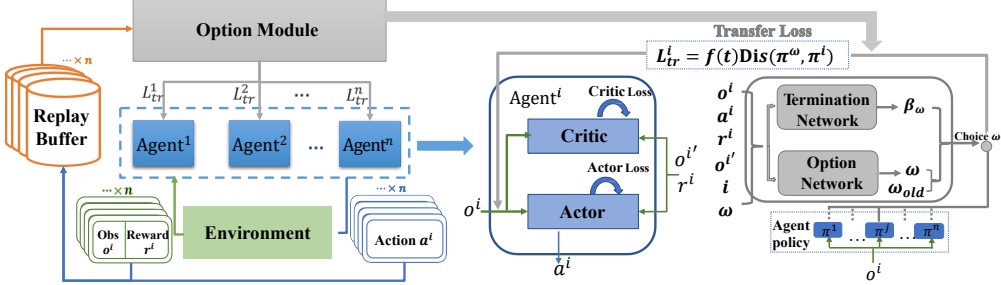

Figure 1: Framework overview.

In this section, we describe our MAPTF in detail. Figure 1 illustrates the MAPTF, which contains two modules, the agent's module with $n$ agents interacting with an environment, and the option module which determines which agent's policy is useful for each agent. The option module first initializes the option set with $n$ options: $\Omega = \{\omega^1, \omega^2, \cdots, \omega^n\}$, each $\omega^i$ is a tuple $\{\mathcal{I}_{\omega^i}, \pi_{\omega^i}, \beta_{\omega^i}\}$ and $\pi_{\omega^i}$ equals to agent $i$'s policy $\pi^i$. At the start of each episode, for each agent, the option module selects an option $\omega$ based on the option-value function and the termination function. Each option terminates according to its termination function and then another option is selected to repeat the process. During the training phase, the option module uses experiences from all agents to update the option-value function and corresponding termination function. Each agent will exploit the knowledge from another agent $\pi^\omega$ based on the selected option $\omega$. This is achieved through policy imitation, which serves as a complementary optimization objective (the option module is responsible for this and each agent does not know which policy it imitates and how the extra loss function is calculated). The exploitation is terminated as the selected option terminates, and then another option is selected to repeat the process. In this way, each agent efficiently exploits useful information from other agents, and as a result, the learning process of the whole system is accelerated and improved.

In the following section, all agents share the same option set based on the assumption that all agents are homogeneous and each agent's policy may be helpful for other agents. MAPTF would be easily

established in the situation where the option module initializes different option sets for each agent, e.g., each agent only needs to imitate a small number of agents. In this case, instead of inputting states into the option-value network and outputting a fixed number of option-values, we input each state-option pair to the network and output a single option-value [23, 25, 11].

## 3.2 MAPTF

---

**Algorithm 1** MAPTF-PPO

---

**Input:** option set $\Omega = \{\omega^1, \omega^2, \cdots, \omega^n\}$, replay buffer $\mathcal{D}^i$, parameters of actor network $\rho^i$ and critic network $\upsilon^i$ for each agent $i$

1: **for** each episode **do**
2:     Select an option $\omega$ for each agent $i$
3:     Select an action $a^i \sim \pi^i(o^i)$ for each agent $i$
4:     Perform $\vec{a}$, observe $\vec{r}$ and new state $s'$
5:     Store transition $(o^i, a^i, r^i, o^{i'}, \omega, i)$ to $\mathcal{D}^i$
6:     Select $\omega'$ if $\omega$ terminates for each agent $i$
7:     **for** each agent $i$ **do**
8:         Optimize the critic loss w.r.t $\upsilon^i$ (Equation 3)
9:         MAPTF calculates the transfer loss $L^i_{tr}$
10:        Optimize the actor loss w.r.t $\rho^i$ (Equation 4)
11:    **end for**
12:    Update the option module (Algorithm 2)
13: **end for**

---

In this section, we describe how MAPTF is applied in PPO [29], a popular single-agent RL algorithm. The way MAPTF combines with other RL and MARL algorithms is similar. The whole process of MAPTF combined with PPO is shown in Algorithm 1. With the input of $n$ options $\Omega = \{\omega^1, \omega^2, \cdots, \omega^n\}$, for each episode, the option module selects an option $\omega$ for each agent (Line 2), and each agent selects an action $a^i$ following its policy $\pi^i$ (Line 3). The joint action $\vec{a}$ is performed, then the reward **r** and new state $s'$ is returned from the environment (Line 4). The transition is stored in each agent's replay buffer $\mathcal{D}^i$ (Line 5). If $\omega$ terminates, then the option module selects another option $\omega'$ for each agent (Line 6).

For each update step, each agent updates its critic network by minimizing the loss $L^i_c$ (Line 8):

$$L^i_c = -\sum_{t=1}^{T} (\sum_{t'>t} \gamma^{t'-t} r^i_t - V_{\upsilon^i}(o^i_t))^2, \tag{3}$$

where $T$ is the length of the trajectory segment in PPO. Then each agent updates its actor network by minimizing the summation of the original loss and the transfer loss $L^i_{tr}$ (Line 10):

$$\bar{L}^i_a = \sum_{t=1}^{T} \frac{\pi^i(a^i_t|o^i_t)}{\pi^i_{old}(a^i_t|o^i_t)} A^i - \lambda KL[\pi^i_{old}|\pi^i] + L^i_{tr}, \tag{4}$$

where $A^i = \sum_{t'>t} \gamma^{t'-t} r^i_t - V_{\upsilon^i}(o^i_t)$ is the advantage function of agent $i$.

To transfer useful knowledge among agents, MAPTF calculates the distance $\text{Dis}(\pi^\omega|\pi^i)$ between each exploited policy $\pi^\omega$ and each agent's policy $\pi^i$, and transfers the loss to each agent respectively, serving as a complementary optimization objective for each agent. This means that apart from maximizing the cumulate reward, each agent also imitates another agent's policy $\pi^\omega$ by minimizing the loss function $L^i_{tr}$ as follows:

$$L^i_{tr} = f(t)\text{Dis}(\pi^\omega|\pi^i), \tag{5}$$

where, $f(t) = 0.5 + \tanh(3 - \mu t)/2$ is the discounting factor. $\mu$ is a hyper-parameter that controls the decreasing degree of the weight. This means that at the beginning of learning, each agent exploits knowledge from other agents mostly. As learning continues, knowledge from other agents becomes less useful and each agent focuses more on the current self-learned policy. We consider the MAPTF is a general framework that can be combined with any existing Deep RL and MARL algorithms.

For policy-based algorithms, the corresponding term is the cross-entropy loss: $H(\pi^\omega | \pi^i)$ (and other choices of the distance metric are also suitable). For value-based algorithms, MAPTF measures the distance of two Q-value distributions.

The next issue is how to update the option module. To evaluate which agent's policy is useful for each agent, the option module needs to collect all agent's experiences for the update. What if the experience from one agent is inconsistent with others? In a POSG, each agent can only obtain the local observation and individual reward signal, which may be different for different agents even at the same state, e.g., each agent has an individual goal to achieve or has different roles, and the rewards assigned to each agent are different. If we use inconsistent experiences to update the same option-value and termination probability, the estimation of the option-value function would oscillate and become inaccurate. To this end, We propose a novel option learning algorithm, the Successor Representation Option (SRO) to address this problem, which is described in the next section.

### 3.3 SRO Learning

MAPTF applies a novel option learning algorithm, Successor Representation Option (SRO) learning to learn the option-value function under each agent's preference. The SRO network architecture is shown in Figure 2, with each observation $o^i$ from each agent $i$ as input. $o^i$ corresponding to the global state $s$ is inputted through two fully-connected layers to generate the state embedding $\phi_{o^i}$, which is transmitted to three network sub-modules. The first sub-module contains the state reconstruction model which ensures $\phi_{o^i}$ well representing $o^i$, and the weights for the immediate reward approximation at local observation $o^i$. The immediate reward is approximated as a linear

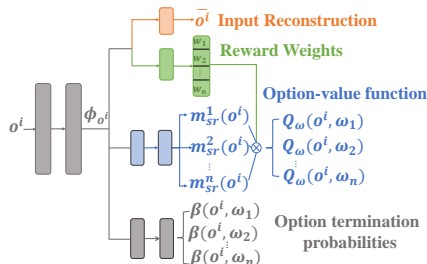

Figure 2: The SRO architecture.

function of $\phi_{o^i}$: $\mathcal{R}^i(\phi_{o^i}) \approx \phi_{o^i} \cdot \mathbf{w}$, where $\mathbf{w} \in \mathbb{R}^D$ is the weight vector. The second sub-module is used to approximate SR for options $m_{sr}(\phi_{o^i}, \omega | \tau)$ which describes the expected discounted future state occupancy of executing the option $\omega$. The last sub-module is used to update the termination probability $\beta(\phi_{o^{i'}}, \omega | \varpi)$.

Given $m_{sr}(\phi_{o^i}, \omega | \tau)$, the SRO-value function can be approximated as: $Q_\omega(\phi_{o^i}, \omega) \approx m_{sr}(\phi_{o^i}, \omega | \tau) \cdot \mathbf{w}$. Since options are temporal abstractions [30], SRO also needs to calculate the $\tilde{U}$ function which is served as SRO upon arrival, indicating the expected discounted future state occupancy of executing an option $\omega$ upon entering a local observation $o^{i'}$:

$$\tilde{U}(\phi_{o^{i'}}, \omega | \tau') = (1 - \beta(\phi_{o^{i'}}, \omega | \varpi)) m_{sr}(\phi_{o^{i'}}, \omega | \tau') + \beta(\phi_{o^{i'}}, \omega | \varpi) m_{sr}(\phi_{o^{i'}}, \omega' | \tau'), \tag{6}$$

where $\omega' = \operatorname{argmax}_{\omega \in \Omega} m_{sr}(\phi_{o^{i'}}, \omega | \tau') \cdot \mathbf{w}$.

The learning process of SRO is shown in Algorithm 2. It first initialized the network parameters for the SRO network and the target network. Then, for each update step, it samples a batch of $B/N$ transitions from each agent's buffer $\mathcal{D}^i$, which means there are $B$ transitions in total (Line 2). SRO loss is composed of three components: the state reconstruction loss $L(\bar{\theta}, \theta)$, the loss for reward weights $L(\mathbf{w}, \theta)$ and SR loss $L(\tau, \theta)$. The state reconstruction network is updated by minimizing two losses $L(\bar{\theta}, \theta)$ and $L(\mathbf{w}, \theta)$ (Lines 3,4):

$$L(\bar{\theta}, \theta) = \left(g_{\bar{\theta}}(\phi_{o^i}) - o^i\right)^2,$$
$$L(\mathbf{w}, \theta) = \left(r^i - \phi_{o^i} \cdot \mathbf{w}\right)^2. \tag{7}$$

The second sub-module, SR network approximates SRO and is updated by minimizing the standard L2 loss $L(\tau, \theta)$ (Lines 5-8):

$$L(\tau, \theta) = \frac{1}{B} \sum_b \left(y_b - m_{sr}(\phi_{o^i}, \omega | \tau)\right)^2, \tag{8}$$

where $y_b = \phi_{o^i} + \gamma \tilde{U}(\phi_{o^{i'}}, \omega | \tau)$. At last, the termination probability of the selected option is updated. According to the call-and-return option execution model, the termination probability $\beta_{\bar{\varpi}}$ controls

**Algorithm 2** SRO Learning.

**Input:** option set $\Omega = \{\omega^1, \omega^2, \cdots, \omega^n\}$, parameters of state feature $\theta$, reward weights $\mathbf{w}$, state reconstruction $\bar{\theta}$, termination network $\varpi$, SR network $\tau$, SR target network $\tau'$; replay buffer $\mathcal{D}^i$ for each agent $i$;

1: **for** each update step **do**
2:    Select $B/N$ samples $(o^i, a^i, r^i, o^{i'}, \omega, i)$ from each $\mathcal{D}^i$
3:    Optimize $L(\bar{\theta}, \theta)$ w.r.t $\bar{\theta}, \theta$ (Equation 7)
4:    Optimize $L(\mathbf{w}, \theta)$ w.r.t $\mathbf{w}, \theta$ (Equation 7)
5:    **for** each $\omega$ **do**
6:      **if** $\pi^\omega$ selects action $a^i$ at observation $o^i$ **then**
7:         Calculate $\tilde{U}(\phi_{o^{i'}}, \omega | \tau')$ (Equation 6)
8:         Optimize $L(\tau, \theta)$ w.r.t $\tau$ (Equation 8)
9:         Optimize the termination network w.r.t $\varpi$ (Equation 9)
10:     **end if**
11:   **end for**
12:   Copy $\tau$ to $\tau'$ every $k$ steps
13: **end for**

when to terminate the selected option and then to select another option accordingly, which is updated w.r.t $\varpi$ as follows (Line 9):

$$\varpi = \varpi - \alpha_\varpi \frac{\partial \beta(\phi_{o^{i'}}, \omega | \varpi)}{\partial \varpi} \left( A(\phi_{o^{i'}}, \omega | \tau') + \xi \right), \tag{9}$$

where $A(\phi_{o^{i'}}, \omega | \tau')$ is the advantage function and approximated as $m_{sr}(\phi_{o^{i'}}, \omega | \tau') \cdot \mathbf{w} - \max_{\omega \in \Omega} m_{sr}(\phi_{o^{i'}}, \omega | \tau') \cdot \mathbf{w}$, and $\xi$ is a regularization term to ensure explorations [2, 36]. At last, the target network parameterized by $\tau'$ copies from $\tau$ every $k$ steps (Line 12).

## 4 Experimental Results

We evaluate the performance of MAPTF combined with the popular single-agent RL algorithm (PPO [29]) and MARL algorithm (MADDPG [25] and QMIX [28]) on two representative multiagent games, Pac-Man [31] and multiagent particle environment (MPE) [25] (illustrated in the appendix). Specifically, we first combine MAPTF with PPO on Pac-Man to validate whether MAPTF successfully solves the sample inconsistency due to the partial observation. Then, we combine MAPTF with three baselines (PPO, MADDPG and QMIX) on MPE to further validate whether MAPTF is a more flexible way for knowledge transfer among agents. We also compare with DVM [32], which is a recent multiagent transfer method. All results are averaged over 10 seeds. More experimental details and parameters settings are detailed in the appendix, source code is provided on `https://github.com/tianpeiyang/MAPTF_code`.

### 4.1 Pac-Man

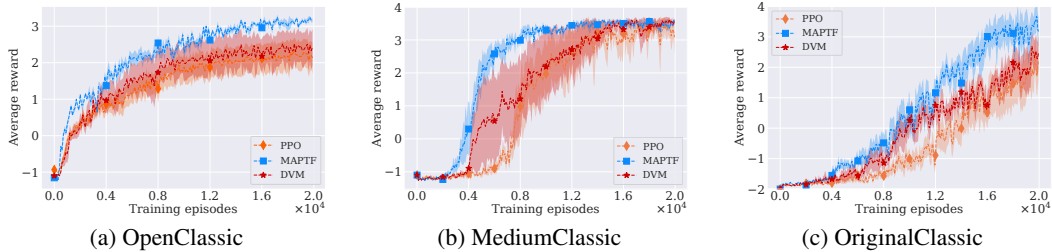

(a) OpenClassic          (b) MediumClassic          (c) OriginalClassic

Figure 3: The performance on Pac-Man under PPO.

Pac-Man [31] is a mixed cooperative-competitive maze game with one pac-man player and several ghost players. We consider three Pac-Man scenarios (OpenClassic, MediumClassic, and Original-

Classic) with the game difficulties increasing. The goal of the pac-man player is to eat as many pills as possible and avoid the pursuit of ghost players. For ghost players, they aim to capture the pac-man player as soon as possible. In our settings, MAPTF controls several ghost players to catch a pac-man player controlled by a well pre-trained PPO policy. The game ends when one ghost catches the pac-man player, or the episode exceeds 100 steps. Each ghost player receives $-0.01$ penalty for each step and a $+5$ reward for catching the pac-man player.

Figure 3 (a) presents the average rewards on the OpenClassic scenario. We can see that MAPTF performs better than other methods and achieves the average discount rewards of $+3$ approximately with a smaller variance. In contrast, PPO and DVM only achieve the average discount rewards of $+2.5$ approximately with a larger variance. This phenomenon indicates that MAPTF enables efficient knowledge transfer between ghost players, thus facilitating better performance.

Next, we consider a complex scenario with a larger layout size than the former, and it contains obstacles (walls). Figure 3 (b) shows the advantage of MAPTF is much more apparent compared with PPO and DVM. Furthermore, MAPTF performs best among all methods, which means it effectively recognizes more useful information for each agent. MAPTF performs better than DVM because MAPTF enables each agent to effectively exploit useful information from other agents, which successfully avoids negative transfer when other agents' policies are only partially useful. However, DVM just transfers all information from other agents through policy distillation without distinction.

Finally, we consider a scenario with the largest layout size, and four ghost players catching one pac-man player. Similar results can be observed in Figure 3 (c). By comparing the results of the three scenarios, we see that the superior advantage of MAPTF increases when faced with more challenging scenarios. Intuitively, as the environmental difficulties increase, agents are harder to explore the environment and learn the optimal policy. In such a case, agents need to exploit other agents' knowledge more efficiently, which would significantly accelerate the learning process, as demonstrated by MAPTF.

## 4.2 MPE

MPE [25] is a multiagent particle world with continuous observation and discrete action space. We consider two scenarios of MPE: predator-prey and cooperative navigation. The predator-prey contains three (nine) agents which are slower and want to catch one (three) adversary (rewarded $+10$ by each hit). The adversary is faster and wants to avoid being hit by the other three (nine) agents. Obstacles block the way. The cooperative navigation contains six (ten) agents to cover six (ten) corresponding landmarks. Agents are penalized with a $-1$ penalty if they collide with other agents. Thus, agents have to learn to cover all the landmarks while avoiding collisions. Both games end when exceeding 100 steps. Both domains contain the sample inconsistency problem since each agent's local observation contains the relative distance between other agents, obstacles, and landmarks. Moreover, in cooperative naviga-

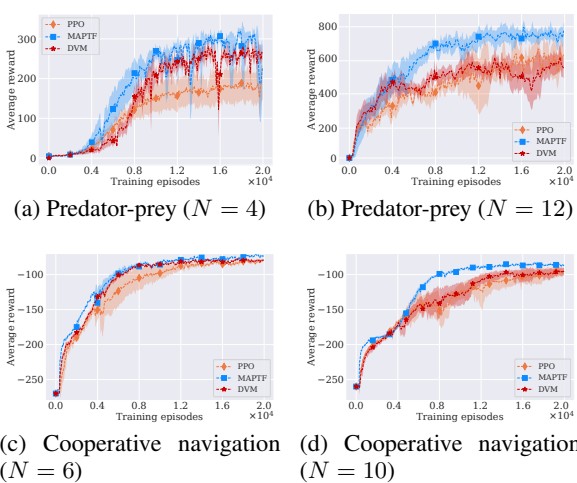

(a) Predator-prey ($N = 4$)   (b) Predator-prey ($N = 12$)

(c) Cooperative navigation ($N = 6$)   (d) Cooperative navigation ($N = 10$)

Figure 4: The performance on MPE under PPO.

tion, each agent is assigned a different task, i.e., approaching a different landmark from others, which means each agent may receive different rewards under the same observation.

Figure 4 (a) shows the average rewards on MPE under PPO. We can see that MAPTF achieves higher average rewards than vanilla PPO and DVM. A similar phenomenon can be found in Figure 4 (b), and the superior advantage of MAPTF is enlarged with the increase in the number of agents. This is because MAPTF successfully solves the sample inconsistency using SRO, and thus efficiently distinguishes which part of the information is useful and provides positive transfer for each agent.

Furthermore, it uses the individual termination probability to determine when to terminate the transfer process, which is more flexible, thus facilitating more efficient and effective knowledge transfer among agents.

Figure 4 (c) and (d) shows the average rewards on cooperative navigation game with six (ten) agents. In this game, agents are required to cover all landmarks while avoiding collisions. We can see that MAPTF performs best among all methods, which means it causes fewer collisions and keeps a shorter average distance from landmarks than other methods. The advantage of MAPTF is due to its effectiveness in identifying useful information from other agents' policies. Therefore, each agent exploits useful knowledge of other agents and, as a result, thus leads to the least collisions and the minimum distance from landmarks.

Finally, we present the performance of MAPTF combined with MADDPG and QMIX on MPE tasks shown in Figure 5 and Figure 6. To further validate the advantage of SRO, we also provide the results of MAPTF with traditional option learning (denoted as MAPTF w/o SRO in Figure 5). MAPTF w/o SRO contains the option module which learns the option value function following single-agent option learning [2, 36]. We can observe that MAPTF performs best among all methods. Although MAPTF with traditional option learning performs better than MADDPG and QMIX learning from scratch, it cannot handle the sample inconsistency problem, thus achieving lower performance than the full MAPTF.

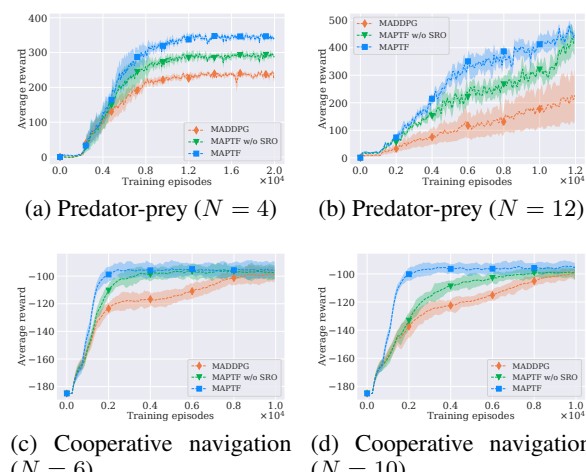

(a) Predator-prey ($N = 4$)  (b) Predator-prey ($N = 12$)

(c) Cooperative navigation ($N = 6$)  (d) Cooperative navigation ($N = 10$)

Figure 5: The performance on MPE under MADDPG.

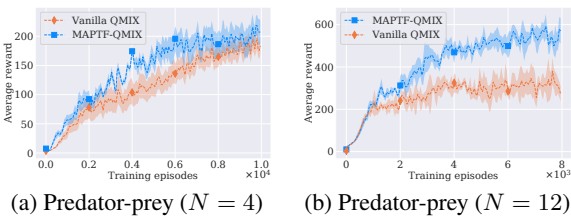

(a) Predator-prey ($N = 4$)  (b) Predator-prey ($N = 12$)

Figure 6: The performance on MPE under QMIX.

## 4.3 Ablation Study

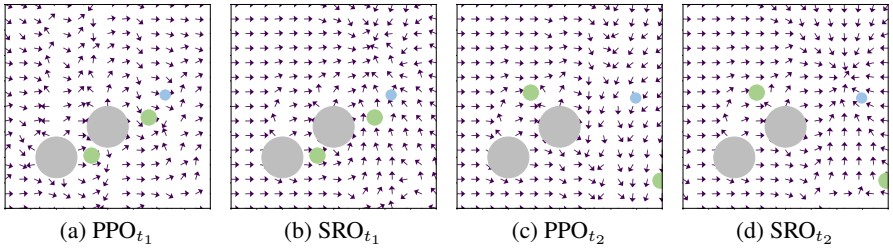

(a) PPO$_{t_1}$    (b) SRO$_{t_1}$    (c) PPO$_{t_2}$    (d) SRO$_{t_2}$

Figure 7: Movements following agent 1's policy and SRO's policy at different timesteps.

**The influence of SRO.** In this section, we first provide an ablation study to investigate whether SRO selects a suitable policy for each agent, thus efficiently enabling agents to exploit useful information from others. Figure 7 presents the action movement in the environment. Each arrow is the direction of movement caused by the specific action at each location. Four figures show the direction of movement caused by the action selected from the policy of an agent at $t_1 = 6 \times 10^5$ steps (Figure 7(a), top left), and at $t_2 = 2 \times 10^6$ (Figure 7(c), bottom left); the direction of movement caused by the action selected from the intra-option policies of SRO at $t_1 = 6 \times 10^5$ steps (Figure 7(b), top right), and at $t_2 = 2 \times 10^6$ steps (Figure 7(d), bottom right) respectively. The preferred direction of movement

should be towards the blue circle. We can see that actions selected by the intra-option policies of SRO are more accurate than those selected from the agent's own policy, namely, more prone to pursue the adversary (blue). This shows that the policy selected by SRO performs better than the agent itself, which means SRO successfully distinguishes useful knowledge from other agents. Therefore, the agent can learn faster and better after exploiting knowledge from this selected policy by SRO than learning from scratch.

**The influence of parameter sharing (PS).** Finally, we investigate the influence of PS, a common trick in multiagent learning, to validate that the superior performance of MAPTF cannot be achieved by PS only. All above MAPTF-PPO experiments do not incorporate PS (both MAPTF-MADDPG and MAPTF-QMIX use PS since we reuse the source code of previous work[25, 28].) Results of the influence of PS on the performance of MAPTF are shown in Figure 8. At the beginning of the training, comparing MAPTF w/ and w/o PS (PPO w/ and w/o PS), PS shows some acceleration since it requires a smaller number of parameters to be updated. However, it does not provide advantages as the training continues. We can see that both MAPTF w/ and w/o PS outperform PPO w/ PS, which validates that the advantage of MAPTF is significant, and its superior performance cannot be achieved by the parameter sharing technique only.

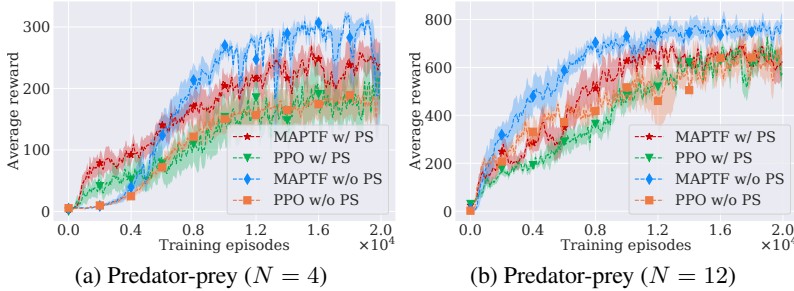

(a) Predator-prey ($N = 4$)        (b) Predator-prey ($N = 12$)

Figure 8: The influence of PS on the performance of MAPTF in predator-prey.

## 5 Related Work

**Tranfer Learning** in MultiAgent Reinforcement Learning (MARL) [6, 17, 4, 16, 7, 8] has been studied with two primary directions: one direction of works focuses on transferring knowledge across multiagent tasks to accelerate the learning process. For example, a number of works explicitly compute the similarities between states or temporal abstractions [18, 3, 10] to transfer across multiagent tasks. The other direction of works transfers knowledge among multiple agents in the same task, which is still investigated at an initial stage. For example, Omidshafiei et al. [27] proposed LeCTR to learn to teach in a multiagent environment. Later, they extended peer-to-peer teaching to a hierarchical structure [19] to improve the teacher credit assignment when faced with long-horizons and delayed rewards problems. However, both LeCTR and HMAT only consider a two-agent scenario.

There are also some works considering to share information among more than two agents, but they suffer from either of the following drawbacks. For example, Liang and Li [22] proposed an attentional teacher-student method under the teacher-student framework where each agent asks for advice from other agents through learning an attentional teacher selector. However, it merely uses the difference of two modes' (self-learning mode and teaching mode) value functions as the reward to train the student policy, which can be very unstable since the value functions are not bounded. Wadhwania et al. [32] proposed a multiagent policy distillation framework to distill a policy from all agents' policies and replace each agent's policy every $k$ steps with it. Similarly, Xue et al. [35] proposed a new algorithm called LTCR to share information among agents through model distillation. However, both DVM and LTCR simply decompose the training process into two stages (i.e., the learning phase and the transfer phase) by turns, and treat all agents without distinction, which may cause negative transfer.

**The option framework** was firstly proposed in [30] as a kind of temporal abstraction which is modeled as Semi-MDPs. A number of works focused on option discovery in single-agent RL

[2, 20, 14, 15]. An important example is the option-critic [2] which learns multiple source policies in the form of options from scratch, end-to-end. However, the option-critic tends to collapse to single-action primitives in later training stages. Later, several works are proposed to overcome this problem[14, 15]. There are also some works following this direction and use options in MARL settings, such as Macro-Action-based MARL[34], dynamic termination options [12] and DOC [5]. These works learn to solve Dec-PoMDP problems using options. The objective of all these works and MAPTF are orthogonal, that MAPTF transfers knowledge among agents and the rest of works learn the policy from scratch, which is not the focus of this work.

## 6 Discussion

We developed a general Multiagent Policy Transfer Framework (MAPTF) for multiagent learning. As noted, the knowledge transfer among agents is achieved through policy imitation. This complementary objective needs to calculate the distance between two agents' policies, as well as the weighting factor $f(t) = 0.5 + \tanh(3 - \mu t)/2$. $\mu$ controls the decreasing degree of the weight is important for our work because an unfavorable value of $\mu$ may cause negative transfer. Currently, we empirically set the value of $\mu$ for different multiagent tasks (detailed in the appendix). How to automatically adjust this parameter to relax the restriction of our work leaves for future work.

Perhaps the biggest remaining limitation of our work is that MAPTF does not make a big contribution to multiagent coordination which is an important problem in MASs. MAPTF learns which agent's policy is useful for each agent from the perspective of each agent's local view, other than the global view. which may be stuck in local optimal in some cases. An effective way to achieve coordination among agents is credit assignment. So a natural extension of MAPTF is to design the option module in a standard centralized training, decentralized execution manner, i.e., learning the joint option-value function and then decomposing it into individual ones and updating each individual option's termination function separately. As a result, decisions are made with respect to both local and global perspectives. The precise description and formulation of this extension, as well as the training and testing, are left for future work.

## 7 Conclusion and Future Work

In this paper, we propose a novel Multiagent Policy Transfer Framework (MAPTF) for efficient MARL by taking advantage of knowledge transfer among agents. MAPTF models the knowledge transfer among agents as the option learning problem to determine which agent's policy is useful for each agent, and when to terminate it. Furthermore, to resolve the sample inconsistency problem, we propose the Successor Representation Option learning, which decouples the environment dynamics from rewards to learn the option-value function under each agent's preference. MAPTF can be easily combined with existing DRL and MARL approaches to significantly boost their performance, as shown by the experimental results. How to achieve multiagent coordination in more complex multiagent problems leaves our future work.

## Acknowledgements

The work is supported by the National Natural Science Foundation of China (Grant No.: U1836214) and the new Generation of Artificial Intelligence Science and Technology Major Project of Tianjin under grant: 19ZXZNGX00010.

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
