## A   Environment illustrations and descriptions

Pac-Man [2] is a mixed cooperative-competitive maze game with one pac-man player and several ghost players (Figure 1). We consider three pac-man scenarios containing two scenarios (OpenClassic (Figure 1 (a)) and MediumClassic (Figure 1 (b))) with two ghost players and one pac-man player and the complex scenario (Figure 1 (c)) with four ghost players and one pac-man player. The pac-man player's goal is to eat as many pills (denoted as white circles in the grids) as possible and avoid the pursuit of ghost players. For ghost players, they aim to capture the pac-man player as soon as possible. In our settings, we aim to control ghost players and the pac-man player is the opponent

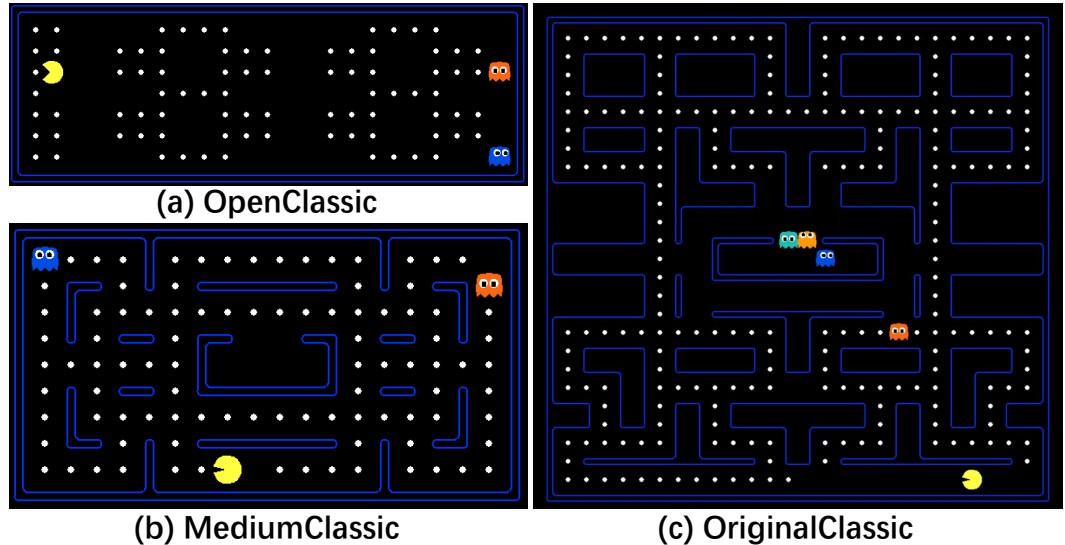

(a) OpenClassic

(b) MediumClassic

(c) OriginalClassic

Figure 1: Pac-Man.

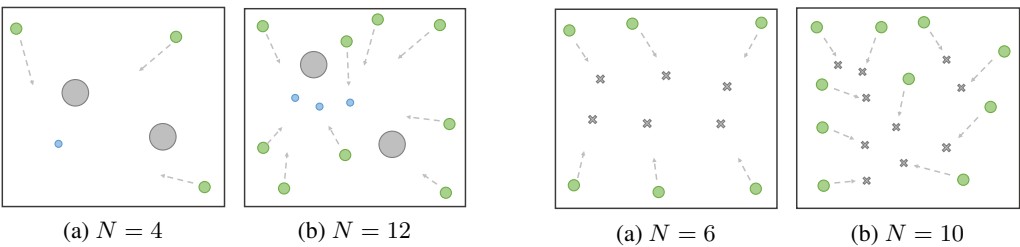

(a) $N = 4$      (b) $N = 12$

Figure 2: Predator-prey.

(a) $N = 6$      (b) $N = 10$

Figure 3: Cooperative Navigation.

controlled by a well pre-trained PPO policy. The game ends when one ghost catches the pac-man player or the episode exceeds 100 steps. Each ghost player receives $-0.01$ penalty for each step and $+5$ reward for catching the pac-man player.

MPE [1] is a multiagent particle world with continuous observation and discrete action space. We choose two scenarios of MPE: predator-prey (Figure 2), and cooperative navigation (Figure 3). The predator-prey contains three (nine) agents (green) which are slower and want to catch one (three) adversaries (blue) (rewarded $+10$ by each hit). Adversaries are faster and want to avoid being hit by the other three (nine) agents. Obstacles (grey) block the way. The cooperative navigation contains six (ten) agents (green), and six (ten) corresponding landmarks (cross). Agents are penalized with a reward of $-1$ if they collide with other agents. Thus, agents have to learn to cover all the landmarks while avoiding collisions. At each step, each agent receives a reward of the negative value of the distance between the nearest landmark and itself. Both games end when exceeding 100 steps.

**State Description**

**Pac-Man** The layout size of two scenarios are $25 \times 9$ (OpenClassic), $20 \times 11$ (MediumClassic) and $28 \times 27$ (OriginalClassic) respectively. The observation of each ghost player contains its position, the position of its teammate, walls, pills, and the pac-man, which is encoded as a one-hot vector. The input of the network is a 68-dimension in OpenClassic, 62-dimension in MediumClassic and 111-dimension in OriginalClassic.

**MPE** The observation of each agent contains its velocity, position, and the relative distance between landmarks, blocks, and other agents, which is composed of 18-dimension in predator-prey with four

agents (36-dimension with twelve agents), 36-dimension with six agents (60-dimension with ten agents) as the network input.

# B  Network structure and parameter settings

The experiments are conducted on a device with CPU of 64 cores, GPU of RTX2080TI and 256G Memory.

**Network Structure** Here we provide the network structure for PPO, MADDPG, QMIX and MAPTF respectively. 1) PPO: for each agent $i$, the actor network has two fully-connected hidden layers both with 64 hidden units, the output layer is a fully-connected layer that outputs the action probabilities for all actions; the critic network contains two fully-connected hidden layers both with 64 hidden units and a fully-connected output layer with a single output: the state value;

2) MADDPG: the actor network has two fully-connected hidden layers, one with 128 hidden units, the second layer with 64 hidden units; the output layer is a fully-connected layer that outputs one single action; the critic network contains two fully-connected hidden layers, one with 128 hidden units, the second layer with 64 hidden units; and a fully-connected output layer with a single output: the state-action value;

3) QMIX: for each agent $i$, the Q network has two fully-connected hidden layers, both with 128 hidden units; the output layer is a fully-connected layer that outputs the Q-values for all actions; the mixing network contains two hypernetworks with 128 hidden units a mixing layer with 32 hidden units; and a fully-connected output layer with a single output: the joint state-action value;

4) SRO network structure is provided in Figure 4.

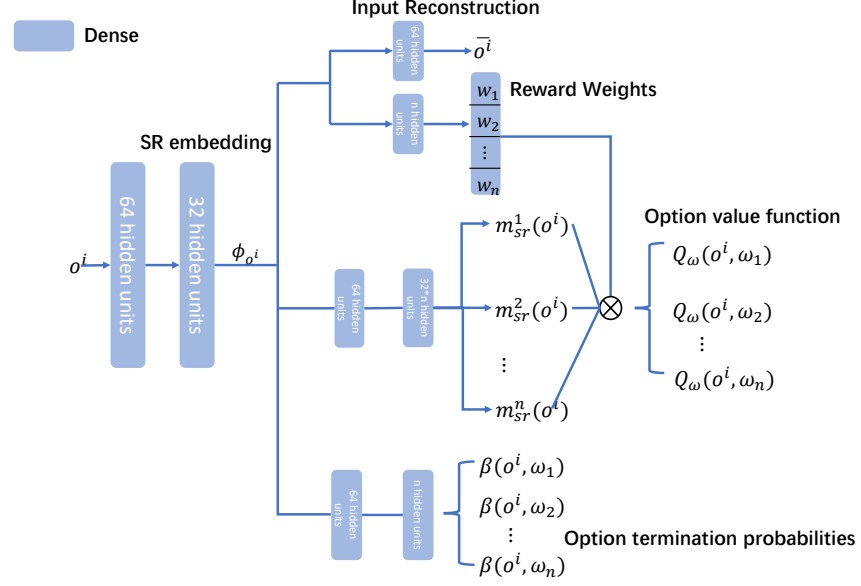

Figure 4: Network structures.

**Parameter Settings**

Here we provide the hyperparameters for MAPTF, DVM as well as three baselines, PPO, MADDPG and QMIX shown in Table 1, 2 and 3 respectively.

Table 1: Hyperparameters for all methods based on PPO.

| Hyperparameter | Value |
|---|---|
| Learning rate | $3e-4$ |
| Length of trajectory segment $T$ | 32 |
| Gradient norm clip $\lambda$ | 0.2 |
| Optimizer | Adam |
| Discount factor $\gamma$ | 0.99 |
| Batch size $B$ of the option module | 32 |
| Replay memory size | $1e5$ |
| Learning rate | $1e-5$ |
| $\mu$ | $5e-4$ |
| $\xi$ | $5e-3$ |
| Action-selector | $\epsilon$-greedy |
| $\epsilon$-start | 1.0 |
| $\epsilon$-finish | 0.05 |
| $\epsilon$ anneal time | $5e4$ step |
| target-update-interval $\tau$ | 1000 |
| distillation-interval for DVM | $2e5$ step |
| distillation-iteration for DVM | 2048 step |

Table 2: Hyperparameters for all methods based on MADDPG.

| Hyperparameter | Value |
|---|---|
| Learning rate | $1e-2$ |
| Batch size | 1024 |
| Optimizer | Adam |
| Discount factor $\gamma$ | 0.99 |
| Batch size $B$ of the option module | 32 |
| Replay memory size | $1e5$ |
| Learning rate | $1e-5$ |
| $\mu$ | $5e-4$ |
| $\xi$ | $5e-3$ |
| Action-selector | $\epsilon$-greedy |
| $\epsilon$-start | 1.0 |
| $\epsilon$-finish | 0.05 |
| $\epsilon$ anneal time | $5e4$ step |
| target-update-interval $\tau$ | 1000 |

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

Table 3: Hyperparameters for all methods based on QMIX.

| Hyperparameter | Value |
|---|---|
| Learning rate | $3e-4$ |
| Batch size | 64 |
| Optimizer | Adam |
| Discount factor $\gamma$ | 0.99 |
| $\epsilon$-start | 1.0 |
| $\epsilon$-finish | 0.05 |
| $\epsilon$ anneal time | $5e3$ step |
| Batch size $B$ of the option module | 32 |
| Replay memory size | $1e5$ |
| Learning rate | $1e-5$ |
| $\mu$ | $5e-4$ |
| $\xi$ | $5e-3$ |
| Action-selector | $\epsilon$-greedy |
| $\epsilon$-start | 1.0 |
| $\epsilon$-finish | 0.05 |
| $\epsilon$ anneal time | $5e4$ step |
| target-update-interval $\tau$ | 1000 |