# OpenReview forum: "An Efficient Transfer Learning Framework for Multiagent Reinforcement Learning"
_NeurIPS.cc/2021/Conference — NeurIPS 2021 Poster_

### Official Review · Reviewer_G6LK · 2021-07-01

**Rating:** 6
**Confidence:** 4

**Summary:**

The authors propose a method for accelerating learning in collaborative multiagent RL domains. The acceleration is achieved by the sharing of information between the agents controlled by a framework that also involves learning multiple options. The authors show clear performance improvements in two different domains against "regular" learning and another transfer learning-related method. However, my main concern with this paper is that it is very difficult to read and understand even at a high level. Therefore, I am not sure I would be able to reproduce what is reported here.

**Limitations And Societal Impact:**

No short-term societal impact.

**Main Review:**

Although the authors present clear performance improvements in the experimental evaluation, there are multiple points to improve in regard to clarity and the description of the method.

The first point would be the title, which is overly generic. The authors should make sure that the title reflects what is novel in this paper and the main characteristics of the proposed method. While it is true that it is a transfer learning method, dozens of such methods exist, and this title doesn't help a reader screening for papers.

The manuscript itself is also very hard to read. Why are the options a critical point in the knowledge-sharing mechanism? If all options are learned from scratch, how I can expect that they should be particularly useful for other agents? ALso, it is not clear at all exactly which information is communicated to the other agents and how often communication is performed.

Reading your paper, I feel like it could be useful to choose which option to be applied in each step through an attention layer, similarly as it is done in DECAF:

Glatt, Ruben, et al. "DECAF: deep Case-based Policy Inference for knowledge transfer in Reinforcement Learning." Expert Systems with Applications 156 (2020): 113420.

The DVM  algorithm is used for comparisons in the experimental evaluation, but you never explained why this method is most similar to yours, and how they differ (in fact, I think not even the acronym is explained). It would also be very relevant to show your comparisons how often the agents communicate. Communication is a critical aspect in any multiagent application, and it would be good to show the trade-off performance improvement x communication used.

I don't see any reason to show the performance for PPO and a multiagent base method for the same domain. One should be fine.

Finally, I am concerned that both of the domains chosen in your evaluation do not require coordination (i.e., the agents can probably solve the task by simply chasing pac-man/prey independently). It would be good to include a domain where coordination is required.

**Time Spent Reviewing:**

2

---

> ### Author Response · Authors · 2021-08-10
> **We thank the reviewer for your inspiring and useful feedback. Please see the response below.**
>
> We thank the reviewer for your inspiring and useful feedback. Please see the response below.
>
> [Title] Thanks for your advice, we will consider a more meaningful name for this paper.
>
> [Options are the critical point] The significance of the option module is validated in section 4.3, although the SRO and each agent’s policies learn simultaneously, SRO identifies which agent’s policy is better at a specific state, so it performs better than a single agent’s policy.
>
> [Communication] There is no direct communication between agents. The knowledge is transferred by the option module during the training phase. Then each agent executes its own policy independently. We think this operation does not violate the MARL training setting.
>
> [Comparison on various methods] We regard our MAPTF as a general framework which can be combined with existing RL and MARL methods, so we show the results of MAPTF combined with PPO and MADDPG, (we also provided results of MAPTF combined with QMIX, please see the response to Reviewer 1). Thanks for pointing out this and the DECAF paper, we will consider using an attention layer to determine the option selection phase as a future direction.
>
> [DVM] We only mention DVM is a recent multiagent transfer work, not a similar work to ours. Both DVM and our MAPTF use knowledge distillation during the training phase. Compared with DVM, our MAPTF establishes a fine-grained knowledge transfer procedure, and we use SRO to identify the useful knowledge based on each agent’s preference. Specifically, DVM distills all agents’ policies every 2000 episodes, and the distillation proceeds 1000 episodes.
>
> [Selected domains] As described in section 6, this paper does not address multiagent coordination deeply. But our selected domains require multiagent coordination, such as predator and prey, the received reward is different when a different number of predators catching the prey. Therefore, if agents behave cooperatively, the prey would be caught by more predators, which causes larger rewards.

---

> ### Author Response · Authors · 2021-09-01
> **Resolving any pending concerns**
>
> We would really appreciate it if the reviewer can let us know if they have any pending concerns and if our response addressed some/all of their concerns. We will try to address them before the discussion period ends.

---

> > ### Comment · Reviewer_G6LK · 2021-09-01
> > **ok**
> >
> > I already have all the information I needed for the discussion phase. Thanks.

---

### Official Review · Reviewer_q2HS · 2021-07-17

**Rating:** 5
**Confidence:** 4

**Summary:**

The authors propose a Multiagent Policy Transfer Framework (MAPTF) that learns which agent policy is appropriate for reuse and when to terminate it. Consequently, MAPTF is seen as an option learning approach to policy transfer; it uses the Successor Representation Option (SRO) algorithm proposed by the authors to learn the option-value function under each agent's preference.

The originality of the proposal consists of extending the problem of option learning, integrating it with the problem of policy transfer and reuse. Thus, the idea is to help the learning of a particular agent in the MARL context with the execution of other agents' options. MAPTF learns, for each agent, the option-value function and when to terminate the execution of the option. Experiments performed in Pac-Man and in the Multiagent Particle environment (MPE) showed the effectiveness of the proposal.

**Main Review:**

MAPTF belongs to a class of novel combinations of well-known techniques. MAPTF extends the option framework to a MARL context, allowing the reuse of other agents' options in learning. Likewise, the SRO algorithm is an adaptation of previous algorithms – SRO learns the option-value function under the preference of each agent, as well as the termination probability of each option.

The text is a little confusing, thus making it difficult to fully understand the proposal. The evolution of the content given in the Preliminaries section is not perfectly extended to the TL-in-MARL scenario. The notation gets quite confusing.

A question that remains is whether the authors, so as to deal with POSG, simply considered the observation of each agent as state information (i.e, assumed that o = s). Another question is how MARL came about: are they simply agents learning their own policy and completely ignoring the other agents, as if they were completely independent learning? If so, how did one deals with the non-stationarity of the MARL environment? In section 6, the authors comment that they did not address the problem of coordination between agents, but ignoring the non-stationarity of a MARL is more consequential than that. Although the authors claim broad generality for their proposal, I am not convinced that it would work for heterogeneous agents.

The figures could be improved and should be in perfect harmony with both their description in the text and the algorithms presented.

Minor issues:
Algorithm 1, step 2 and 6: How is an option selected? Step 4: How is the new state s’ observed? Step 9: is Eq. 9 used here? How is Dis(.,.) calculated?
Algorithm 2: Is it run for each agent? Make this clear in step 1 of the algorithm.

General question: at the beginning of learning in a MARL, agents still don't know much and exhibit erratic behavior. However, eq. 9 indicates that it uses the policies of other agents the most at the beginning of learning. As everyone is learning at the same time, would the transfer be helpful?

The experimental results presented are very promising. However, as there were many doubts about the modeling carried out, it is not possible to be sure that the contribution is in fact very significant. While the basic ideas are promising, development must be improved.

I would like to reinforce that I have read the authors' rebuttal and thank them for all the clarifications and improvements to the paper.

**Time Spent Reviewing:**

about 4 hours

---

> ### Author Response · Authors · 2021-08-10
> **We thank the reviewer for your inspiring and useful feedback. Please see the response below.**
>
> We thank the reviewer for your inspiring and useful feedback. Please see the response below.
>
> [Preliminaries] Thanks for pointing out this, we have added a formal definition for TL-in MARL in the preliminary section.
>
> [POSG] We follow the POSG setting, the global state $s$ is not simply treated as the local observation $o_i$, there is a mapping function between them. The algorithm procedure for independent PPO combined with MAPTF is independent learning and cannot avoid the non-stationary problem. However, we also combine MADDPG with MAPTF, which learns the agent’s policies in a CTDE manner. We aim to assess the advantage of our framework when it combines with existing MARL or independent RL methods to see whether it boosts these methods.
>
> [Heterogeneous agents] As for heterogeneous agents, one way is to directly share knowledge between agents in the same group using MAPTF. Another direction is to learn abstract common knowledge first and then transfer the knowledge to other agents. This is similar to transfer knowledge among robots with different morphologies in single-agent settings.
>
> [Figure Descriptions] Thanks for pointing out this, we have added the explanations in the caption.
>
> [Minor issues] 1) The option is selected using an $\epsilon$-greedy strategy over the SRO value function. 2) The new state s’ is returned by the environment, but each agent only draws a local observation corresponding to s’. 3) Equation 9 is used in step 9; 4) As described in Lines 188-190, different Dis() is adopted for different MARL methods. 5) Algorithm 2 is run once a step. Thanks for pointing out this, we have revised this to make it clearer.
>
> [General question] As described in our motivation, although each agent’s policy is not learned very well at the beginning of learning in a MAS, the familiarities to different regions of the environment for each agent are different (i.e., each agent’s trajectories cover different regions unevenly). So we aim to use an auxiliary SRO module to transfer knowledge across different agents, which enables agents to know the environment more quickly.

---

> ### Author Response · Authors · 2021-09-01
> **Resolving any pending concerns**
>
> We would really appreciate it if the reviewer can let us know if they have any pending concerns and if our response addressed some/all of their concerns. We will try to address them before the discussion period ends.

---

> > ### Comment · Reviewer_RZaH · 2021-09-02
> > **ACK**
> >
> > Thanks, but I have the information I need and am currently in discussion with other reviewers.

---

### Official Review · Reviewer_RZaH · 2021-07-17

**Rating:** 5
**Confidence:** 4

**Summary:**

The paper proposes a multi-agent algorithm which combines options with a successor representation and imitation learning. My understanding of the algorithm is that it initializes n options (one for each agent), and a meta-policy which selects which option should be used by each agent during each “episode” (see algorithm 1). So agents can use the most relevant option, rather than relying solely on their own option. Individual options are learned with PPO. Successor Representations are used for the meta-policy (I believe) by using a shared feature representation but individual preference vectors for each agent. Finally, the options for all agents are forced to be similar to each other through an imitation learning loss, which is scaled down over the course of training.

**Limitations And Societal Impact:**

The paper provides a nice limitations section that analyzes drawbacks of the method. It does not appear to address societal impact. My suggestion would be to include a discussion of possible beneficial applications of effective multi-agent learning as motivation for why the problem is important.

**Main Review:**

**Clarity:**
- The paper has significant issues with clarity which make it difficult to assess the proposed algorithm. I had to read the methods section several times to parse what is being done.
- For example, consider the following: “The option module select an option ω for each agent. During the training phase, the option module uses experience from all agents to update the option-value function and corresponding termination probabilities.” -> this indicates that there is a single option pool being shared among all agents. This would imply that actually each agent is running the identical policy, which is composed of the same set of options. The paper then goes on to say, “Each agent will exploit the knowledge from another agent πω based on the selected option ω, and this is achieved through policy imitation, which serves as a complementary optimization objective”. This is very confusing, because it is not clear how agents actually have their own policies if they are all using the same option pool to act. How can they imitate each others’ policies if they share the same policy? Only after re-reading this section to determine that there are n options, did I realize that the intention is that each option is meant in some sense to correspond to one agent, although this does not appear to be enforced.
- The second problem is that how the SR is used is not made clear by the paper. A careful reading determines it is used for the option-value function, i.e. the meta-policy. This only became apparent to me after reading that the individual agent policies were learned with PPO, but it took several passes. This should be stated more clearly and earlier in the paper (and repeated).
- A further suggestion to help clarify the paper would be to state explicitly which parameters are unique to each agent. The options are actually not unique to the agents, because they are shared freely among the agents. My current understand is that the only agent-specific parameters are the preference vectors w.
- The motivation given in lines 26-30 is compelling.
- Using \tau for parameters is not a great use of notation, since \tau is often using to represent trajectories in RL.
- The explanation of successor features (lines 87-98) could be made more clear.
- The transfer loss L_{tr} does not appear to be introduced before it is used in Eq. 4
- There are typos and grammar mistakes throughout e.g. “a regularization term to ensure explorations” in line 176

**Novelty/Originality:**
- The novelty of the proposed algorithm is limited. The idea of using options for multi-agent learning has been explored in [33,12,5] as cited in this paper. The idea of using successor representations with a shared feature representation and individual preference vectors for each agent was proposed in https://arxiv.org/abs/2102.12560, which is not cited by this paper.
- The idea of using imitation learning to make the agents’ policies more similar appears to be novel, but it is not well motivated. Why use options at all in this case, rather than allowing agents to share parameters for more efficient learning? This should be better motivated. It would also be good to include an ablation study assessing the importance of this imitation learning loss, and to benchmark against multi-agent techniques which instead rely on parameter sharing among agents.

**Quality:**
- MADDPG is not a reasonable multi-agent baseline. It is several years old (from 2017), and many more performant algorithms have been proposed since then (e.g. QMIX, etc, or https://arxiv.org/pdf/2106.02195.pdf for a thorough collection of SOTA baselines). It is difficult to assess the contributions of this paper unless it is properly benchmarked against recent techniques.
- The paper claims that DVM, published in 2019, is “the most recent multiagent transfer method”. This is not true; see e.g. https://arxiv.org/abs/2102.12560
- The ablation studies do not assess the contribution of the options framework, or the imitation learning approach.
- A positive aspect of the paper is that it uses 10 random seeds per experiment, indicating the results of the experimental evaluations are more reliable and more likely to replicate. Good work!
- The use of the state reconstruction loss to help refine the successor representation is a nice touch.

**Significance:**
- Overall the paper has promise, but it is difficult to assess the significance of the contributions due to the clarity issues and the lack of appropriate benchmarking against recent techniques.

**Time Spent Reviewing:**

2

---

> ### Author Response · Authors · 2021-08-10
> **We thank the reviewer for your inspiring and useful feedback. Please see the response below.**
>
> We thank the reviewer for your inspiring and useful feedback. Please see the response below.
>
> [Clarity]: We are sorry to make these misunderstandings. 1) As described in lines 115-120, in this paper, all agents share the same option set since they are homogeneous, and we assume all other agents’ policies may be helpful. 2)The policy over options is different for each agent since we propose a new algorithm SRO learning to learn the option-value function under each agent’s preference. 3) The SRO is an auxiliary module in our method, and each agent learns its policy and interacts with the environment using its own policy. The SRO value function is used as an indicator to determine at each step, under agent i’s local observation, which agent performs better than agent i, and could be exploited by agent i. We will repolish the paper to make the method section (how to define the option set, how SRO is used, which part of parameters are shared or not) clearer.
>
> [Novelty]: There are some multiagent option methods, as well as a multiagent SR method (which we will cite in the revised version). However, the motivation and methodology in this paper are different. 1) Previous multiagent option methods follow an end-to-end training to use temporal abstractions, i.e., options to learn the optimal policy. Nevertheless, we use options as an auxiliary module to achieve knowledge transfer among multiple agents, learn the option-value functions to determine at each step, under agent i’s local observation, which agent performs better than each agent and could be exploited by the agent. 2) Different from PsiPhi-Learning which decouples the Q(s,a) in the multiagent setting, we extend SR to temporal abstractions and decouple the option-value function Q(s,\omega) to learn the option-value function under each agent's preference, so that the options can identify useful information for each agent, which is one of our main contributions.
>
> [Imitation learning vs parameter sharing and ablation study]: For the baseline method and our method, we both use the parameter sharing technique for each agent’s policy module, as well as the SRO except for parameters of the preference vectors w. So the experiment results show the advantages brought by MAPTF. Thanks for pointing out this, we will conduct an ablation study to assess the contribution of each part of our method.
>
> [Benchmark]: Thanks for pointing out this, we have included the comparison between MAPTF-QMIX vs vanilla QMIX on two scenarios, the convergence speed and performance are shown as follows:
>
>     Scenarios/methods	   Cooperative navigation(N=6)		        Predator-prey(N=12)
>
>                    Convergence Speed      Performance         Convergence Speed     Performance
>
>              (Training episodes)   (Average reward)     (Training episodes)    (Average reward)
>
>     Vanilla QMIX 	2000      	    -82(±4)   	            4500                350(±50)
>
>     MAPTF-QMIX	   1800	          -41(±5)	               4300                443(±30)
>
> As for the paper the reviewer mentioned, this paper is put on ArXiv after the submission deadline, we will cite it in the revised version, and further, consider combining it and other recent MARL methods with our MAPTF.
>
> [Societal impact]: Thanks for pointing out this, we will add this discussion in the revised version.

---

> > ### Comment · Reviewer_RZaH · 2021-08-10
> > **Acknowledgement and further clarifications**
> >
> > Thank you for your response.
> >
> > > [Clarity]: We are sorry to make these misunderstandings. 1) As described in lines 115-120, in this paper, all agents share the same option set since they are homogeneous, and we assume all other agents’ policies may be helpful. 2)The policy over options is different for each agent since we propose a new algorithm SRO learning to learn the option-value function under each agent’s preference. 3) The SRO is an auxiliary module in our method, and each agent learns its policy and interacts with the environment using its own policy. The SRO value function is used as an indicator to determine at each step, under agent i’s local observation, which agent performs better than agent i, and could be exploited by agent i. We will repolish the paper to make the method section (how to define the option set, how SRO is used, which part of parameters are shared or not) clearer.
> >
> > Unfortunately I am still not clear on whether my interpretation of the method is correct. Is it true that the only agent-specific parameters are the preference vectors w?
> >
> > > [Novelty]:
> >
> > Making the differences clear in the related work will strengthen the paper.
> >
> > > [Imitation learning vs parameter sharing and ablation study]:
> >
> > Is there an ablation of MAPTF which includes all components, including SRO, but not the imitation learning loss in Eq. 9?
> >
> > > Thanks for pointing out this, we have included the comparison between MAPTF-QMIX vs vanilla QMIX on two scenarios, the convergence speed and performance are shown as follows:
> >
> > Thank you for including a comparison with QMIX! I believe this baseline significantly strengthens the paper.
> >
> > > As for the paper the reviewer mentioned, this paper is put on ArXiv after the submission deadline, we will cite it in the revised version, and further, consider combining it and other recent MARL methods with our MAPTF.
> >
> > My apologies, I didn't mean that you should compare with the method proposed in https://arxiv.org/pdf/2106.02195.pdf (since it is not highly relevant to this work), but rather that the experimental section of this paper (especially figure 4) can serve as a good example of how to benchmark against relevant SOTA multi-agent methods.

---

> > > ### Author Response · Authors · 2021-08-11
> > > **Thanks for your comments.**
> > >
> > > Thanks for your comments.
> > >
> > > $~$
> > > [Clarity]:
> > > >Yes, each agent maintains specific parameters to learn preference vectors. We are sorry for this, that we only mention this in the response for [Imitation learning vs parameter sharing and ablation study]. And we will add an ablation study about the performance of our MAPTF w/wo parameter sharing to validate the advantage of SRO and imitation learning loss in the revised version.
> > >
> > > $~$
> > > [Novelty]:
> > > >Thanks for pointing out this, we have clarified this in the revised version.
> > >
> > > $~$
> > > [Imitation learning vs parameter sharing and ablation study]:
> > > >The whole framework without the imitation learning loss is no need to learn the SRO value function since the SRO module and the agent module are loosely coupled, the only connection is the imitation learning loss which is calculated by the SRO module and transferred to the agent module. But we will add an ablation study about the performance of our MAPTF w/wo parameter sharing,  w/wo the imitation learning loss to validate the advantage of our method in the revised version.
> > >
> > > We hope that these comments have addressed the reviewer’s concerns about the paper. We are happy to answer any follow-up questions. We thank the reviewer's comments, this helps us a lot to improve the paper!

---

> > > ### Author Response · Authors · 2021-09-01
> > > **Resolving any pending concerns**
> > >
> > > We have added the ablation study the reviewer requested in the general response. We would really appreciate it if the reviewer can let us know if they have any pending concerns and if our response addressed some/all of their concerns. We will try to address them before the discussion period ends.

---

### Decision · Program_Chairs · 2021-09-27

**Decision:**

Accept (Poster)

**Comment:**

The reviewer team agrees that this paper has some clear strengths and weaknesses:
(+) It proposes a new way of combining two known techniques, the option framework and successor representation, for MARL.
(+) The proposed algorithm achieves good performance empirically.
(+) The additional experimental results provided in the author response phase addressed some of the reviewers' concerns, especially the comparison with QMIX and the additional ablation study. These results improve the quality of the paper and the significance of the work.
(-) The applicability of the proposed technique is limited by some level of homogeneity or similarity among the agents. The authors discussed how the technique can be used for heterogeneous agents in the response phase but the benefit of the framework clearly relies on a (sub)set of agents who have similarities in performing the tasks and do not need to carefully coordinate with each other.
(-) While the authors claim that they have made efforts on improving the clarity of the paper since the last submission to an earlier conference, the reviewer team finds that the paper still lacks clarity in a number of places, which makes it hard for the reader to understand the paper. For example, it was not very clear which parameters are shared and which ones are agent-specific. Alg 1 lacks clear explanations in some of the important steps. Some of the related work is mentioned without explaining the differences from this work clearly. The authors' response clarified some of these points but the paper's clarity needs to be improved. In addition, there are typos and grammar mistakes throughout the paper.

Overall, the reviewer team views this paper as a borderline paper mainly due to the lack of clarity. The novel combination of the option framework and successor representation may inspire further research and the significantly improved empirical performance over baselines is promising. In the next version of the paper, I would recommend the authors include the additional experimental results presented in the response phase and improve the clarity of the paper based on reviewers' comments and suggestions.